# Video Prediction Policy:
# A Generalist Robot Policy with Predictive Visual Representations

**Yucheng Hu** [* 1 2]  **Yanjiang Guo** [* 1 3]  **Pengchao Wang** [4]  **Xiaoyu Chen** [1 3]  **Yen-Jen Wang** [5]  **Jianke Zhang** [1 3]
**Koushil Sreenath** [5]  **Chaochao Lu** [2]  **Jianyu Chen** [1 3 4]

## Abstract

Visual representations play a crucial role in developing generalist robotic policies. Previous vision encoders, typically pre-trained with single-image reconstruction or two-image contrastive learning, tend to capture static information, often neglecting the dynamic aspects vital for embodied tasks. Recently, video diffusion models (VDMs) demonstrate the ability to predict future frames and showcase a strong understanding of physical world. We hypothesize that VDMs inherently produce visual representations that encompass both current static information and predicted future dynamics, thereby providing valuable guidance for robot action learning. Based on this hypothesis, we propose the Video Prediction Policy (VPP), which learns implicit inverse dynamics model conditioned on predicted future representations inside VDMs. To predict more precise future, we fine-tune pre-trained video foundation model on robot datasets along with internet human manipulation data. In experiments, VPP achieves a 18.6% relative improvement on the Calvin ABC-D generalization benchmark compared to the previous state-of-the-art, and demonstrates a 31.6% increase in success rates for complex real-world dexterous manipulation tasks. Videos and code are available at `https://video-prediction-policy.github.io`.

## 1. Introduction

Building generalist robot policies capable of solving a variety of tasks is a rapidly advancing area of research (Brohan

---

[*]Equal contribution  [1]IIIS, Tsinghua University  [2]Shanghai AI Lab  [3]Shanghai Qi Zhi Institute  [4]RobotEra  [5]University of California, Berkeley.  Correspondence to: Jianyu Chen <jianyuchen@tsinghua.edu.cn>.

*Proceedings of the $42^{nd}$ International Conference on Machine Learning*, Vancouver, Canada. PMLR 267, 2025. Copyright 2025 by the author(s).

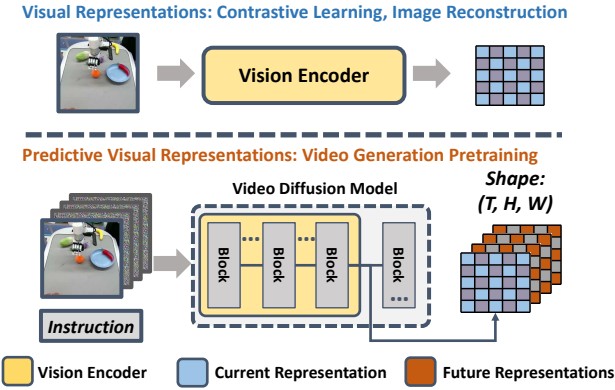

Visual Representations: Contrastive Learning, Image Reconstruction

Predictive Visual Representations: Video Generation Pretraining

*Figure 1.* Visual representations inside video prediction models explicitly express both current and future frames, providing valuable future information for embodied agent. Previous vision encoders did not have explicit future representations.

et al., 2023; Team et al., 2024; Wu et al., 2023a; Guo et al., 2025; Cui et al., 2025; Ding et al., 2024; 2025; Shi et al., 2025; Zhao et al., 2025). A crucial component in these generalist policies is the vision encoder, which captures visual information from pixel observations. Many studies have focused on optimizing vision representations for embodied agents, often leveraging internet video datasets (Ebert et al., 2021; Grauman et al., 2022) and self-supervised techniques such as single-image reconstruction (Majumdar et al., 2023; Karamcheti et al., 2023; Gupta et al., 2024), two-image contrastive learning, and image-text contrastive learning (Nair et al., 2022; Ma et al., 2022). Although these visual pre-training methods have demonstrated success for embodied tasks, they may not fully exploit the dynamic information encoded in sequential video datasets, as they typically operate on only one or two sampled images.

Recently, powerful video diffusion models (VDMs) (Ho et al., 2022; Blattmann et al., 2023a; Hong et al., 2022; Yang et al., 2024) have achieved impressive results in video generation tasks. Instead of performing pre-training operation on single image or pairs of images, VDMs directly model entire video sequences. Text-guided video prediction models (TVPs) (Gu et al., 2023; Chen et al., 2023) can even predict future frames based on current observations

and instructions, demonstrating a good understanding of the physical dynamics.

Inspired by the strong prediction capabilities of TVP models, we hypothesize that they inherently contain valuable physical dynamics knowledge and can produce more effective visual representations for embodied agent. We take a deeper look at the visual representation inside TVP models. These representations are typically structured as a tensor with dimensions $(T, H, W)$, explicitly representing 1 current step and $(T-1)$ predicted future steps (Blattmann et al., 2023a), where $H$ and $W$ correspond to the height and width of the image representation. In contrast, previous vision encoders do not explicitly capture future representations, as shown in Figure 1. Based on this distinction, we refer to these latent variables within the video diffusion model as "predictive visual representations".

Our key insight is that the downstream policy can implicitly learn the inverse dynamics model by tracking the robot's movements within the predictive representation. As long as the video model accurately predicts future scenarios for diverse tasks, the policy can generate appropriate actions by tracking robot arm's position implicitly. In this way, we can transfer the generalization capabilities of the video prediction model to robotic policy. We only need few demonstrations to align the robot's action space with the visual space.

Building on this insight, we introduce the **V**ideo **P**rediction **P**olicy (**VPP**), which employs a two-stage learning process: First, we fine-tune a general-purpose video diffusion model into a text-guided video prediction (TVP) model using internet human and robot manipulation data (Goyal et al., 2017; O'Neill et al., 2023). This step aims to develop a controllable video generation model that improves prediction capabilities in the manipulation domain. In the second stage, we learn a inverse dynamics model conditioned on the predictive representations from the TVP model. Since we direct use the internal representation and avoid the need for multiple denoising steps as in previous work (Black et al., 2023; Du et al., 2024), VPP can operate at high frequency in a closed-loop manner. We also visualize the representations within the VDM and confirm that they effectively capture key information about future evolution.

In experiments, VPP consistently outperform other baseline algorithms across two simulated (Mees et al., 2022; Yu et al., 2020) and two real-world settings, demonstrating the effectiveness of our approach. Notably, the VPP achieves a 41.5% improvement in the Calvin ABC→D benchmark (Mees et al., 2022) compared to the previous SOTA method (Wu et al., 2023a). In real-world experiments, VPP shows a 31.6% improvement in success rate over the strongest baseline on high-dimensional dexterous hand manipulation tasks.

## 2. Related Works

**Visual Representation Learning for Robotics.** Self-supervised learning (SSL) techniques, such as contrastive (Chen et al., 2021; 2020), distillation-based (Baevski et al., 2022; Caron et al., 2020), and reconstructive (He et al., 2022; Bao et al., 2021), have achieved significant advancements in visual representation learning. Prior research has shown that these SSL techniques enable vision encoders to produce effective representations for embodied AI tasks (Yadav et al., 2023b;a; Parisi et al., 2022; Radosavovic et al., 2023; Chen et al., 2024a), capturing both high-level semantic and low-level spatial information. Notably, methods like R3M (Nair et al., 2022), vip (Ma et al., 2022), VC-1 (Majumdar et al., 2023), and Voltron (Karamcheti et al., 2023) have specifically focused on embodied tasks by innovating pre-training approaches on human manipulation video datasets (Goyal et al., 2017; Grauman et al., 2022). However, regardless of the training objective, the learned vision encoders primarily focus on extracting pertinent information from current observations without explicitly predicting future states. In contrast, our Video Prediction Policy leverages predictive representations within video prediction models to explicitly encapsulate both current and predicted future frames.

**Future Prediction for Embodied Control Tasks.** Existing research also explores the use of future prediction to enhance policy learning (Bharadhwaj et al., 2024; Chen et al., 2024b; Ye et al., 2024; Guo et al., 2024; Zhang et al., 2025; Song et al., 2025). For example, SuSIE (Black et al., 2023) conditions its control policy on a predicted future keyframe generated by InstructPix2Pix (Brooks et al., 2023), while UniPi (Du et al., 2024) learns the inverse dynamics between two generated frames. These methods rely on a single future prediction step to determine actions, which may not accurately capture the complexities of physical dynamics. Additionally, they denoise the final future image which is time-cosuming and lead to low control frequency. GR-1 (Wu et al., 2023a) generates subsequent frames and actions autoregressively. However, it only generates one image per forward pass, and its prediction quality lags behind that of diffusion-based methods. Furthermore, GR-1 does not leverage pre-trained video foundation models. In contrast, VPP leverages representation fine-tuned from video foundation model, and predict a sequence of future frames to more effectively inform policy learning.

**Visual Representation inside Diffusion Models.** Diffusion models have achieved remarkable success in the image and video generation tasks (Rombach et al., 2022; Blattmann et al., 2023a). Although diffusion models are trained as denoisers, researches have shown that **image diffusion models** can also function effectively as vision encoders, generating meaningful visual representations that is linear-separable

for discrimination tasks (Xiang et al., 2023) and invaluable for semantic segmentation (Luo et al., 2024). Gupta et al. (2024) also point out that representation inside image diffusion are versatile for embodied tasks. However, the capabilities of representations within **video diffusion models** have not been extensively explored. He et al. (2024) try to use latent representation inside discrete VDMs to assist policy learning, however it need not leverage pre-trained video foundation models and train from scratch. Our findings suggest that representation within pretrained VDMs have a unique predictive property, making them especially useful for sequential embodied control tasks.

## 3. Preliminaries

**Video Diffusion Models.** The core idea of diffusion models is to continuously add Gaussian noise to make video sequences a Gaussian and leverage the denoising process for generating videos. Let $x_0$ represent a real video sample, the forward process aims to add Gaussian noise and result in a set of noisy data, i.e., $q(x_t|x_{t-1}) = \mathcal{N}(x_t; \sqrt{\alpha_t}x_{t-1}, (1 - \alpha_t)\mathbb{I})$, where $x_t$ and $\alpha_t$ indicate the noisy data and noise amplitude at the timestep $t$. Let $\bar{\alpha}_t = \prod_{i=1}^{t} \alpha_i$, the above process can be simplified as:

$$x_t = \sqrt{\bar{\alpha}_t}x_0 + \sqrt{1 - \bar{\alpha}_t}\epsilon_t. \tag{1}$$

The reverse process starts from the most noisy sample $x_T$ can be described in a variational approximation of the probabilities $q(x_{t-1}|x_t)$, as follows:

$$p(x_{t-1}|x_t) = \mathcal{N}(x_{t-1}; \sqrt{\bar{\alpha}_{t-1}}\mu_\theta(x_t, t), (1 - \bar{\alpha}_{t-1})\mathbb{I}). \tag{2}$$

where $\mu_\theta(x_t, t) = (x_t - \sqrt{1 - \bar{\alpha}_t}\epsilon_\theta(x_t, t))/\sqrt{\bar{\alpha}_t}$ is a learnable neural network to estimate $x_{t-1}$. Further, in text-guided video generation, the denoising process learns the noise estimator $\epsilon_\theta(x_t, c)$ to approximate the score function $\sqrt{1 - \bar{\alpha}_t}\nabla_{x_t} \log p_\psi(x_t|c)$, controlling the video generation based on the initial frame and language prompt.

**Diffusion Policy.** The diffusion model has also proven effective in action learning, known as diffusion policy (Chi et al., 2023). The diffusion policy aims to denoise the action sequence $a_i = (\hat{a}_i, \hat{a}_{i+1}, ..., \hat{a}_{i+m})$ based on observations $s_i$ and instruction. Chi et al. (Chi et al., 2023) point out that diffusion policy is capable of expressing complex multimodal action distributions and stabilizing training. Recent work (Reuss et al., 2024) further enhances the diffusion policy by incorporating the advanced diffusion transformer (DiT) block (Peebles & Xie, 2023), a technique we also adopt in the Video Prediction Policy to improve performance.

## 4. Video Prediction Policy

In this section, we describe the two-stage learning process of the Video Prediction Policy, shown in Figure 2. Initially, we train the Text-guided Video Prediction (TVP) model across diverse manipulation datasets to harness physical knowledge from internet data; subsequently, we design networks to aggregate predictive visual representations inside the TVP model and output final robot actions.

### 4.1. Text-guided Video Prediction (TVP) Model for Robot Manipulation.

Recent advancements have focused on training general video generation models using extensive online video datasets, which encode abundant prior knowledge about the physical world's dynamics. However, we notice that these models are not fully controllable and fail to yield optimal results in specialized domains such as robot manipulation. To address this, we fine-tune the general video generation model into a specialized "Manipulation TVP Model" to enhance prediction accuracy.

We chose the open-sourced Stable Video Diffusion (SVD) model (Blattmann et al., 2023a) with 1.5 billion parameters as our foundation. we observe that the open-sourced SVD model conditions only on initial-frame images $s_0$. We augment the model to incorporate CLIP (Radford et al., 2021) language feature $l_{emb}$ using cross-attention layers. Furthermore, we adjust the output video resolution to 16×256×256 to improve training and inference efficiency. Despite these modifications, we preserve the other components of the original pre-trained SVD framework to retain its core capabilities. We denote this modified version as $V_\theta$. In this setup, the initial observation $s_0$ is concatenated channel-wise with each predicted frame as a condition. Then model $V_\theta$ is trained with diffusion objective, reconstructing the full video sequence $x_0 = s_{0:T}$ in dataset $D$ from noised samples $x_t = \sqrt{\bar{\alpha}_t}x_0 + \sqrt{1 - \bar{\alpha}_t}\epsilon$:

$$\mathcal{L}_D = \mathbb{E}_{x_0 \sim D, \epsilon, t} \|V_\theta(x_t, l_{emb}, s_0) - x_0\|^2 \tag{3}$$

The video prediction objective offers a unified interface that directly generates future visual sequences, enabling the TVP model to harness physical knowledge from diverse datasets. These include internet human manipulation datasets $D_H$, internet robot manipulation data $D_R$, and also self-collected datasets $D_C$. Given the varying quality and scale of these datasets, we introduce specific coefficients $\lambda$ to appropriately balance the influence of different dataset types:

$$\mathcal{L}_{video} = \lambda_H \mathcal{L}_{D_H} + \lambda_R \mathcal{L}_{D_R} + \lambda_C \mathcal{L}_{D_C} \tag{4}$$

Then we froze the fine-tuned manipulation TVP models in downstream action learning.

### 4.2. Action Learning Conditioned on Predictive Visual Representation

**TVP Model as Vision Encoder.** After training the TVP model specifically for manipulation tasks, it can accurately

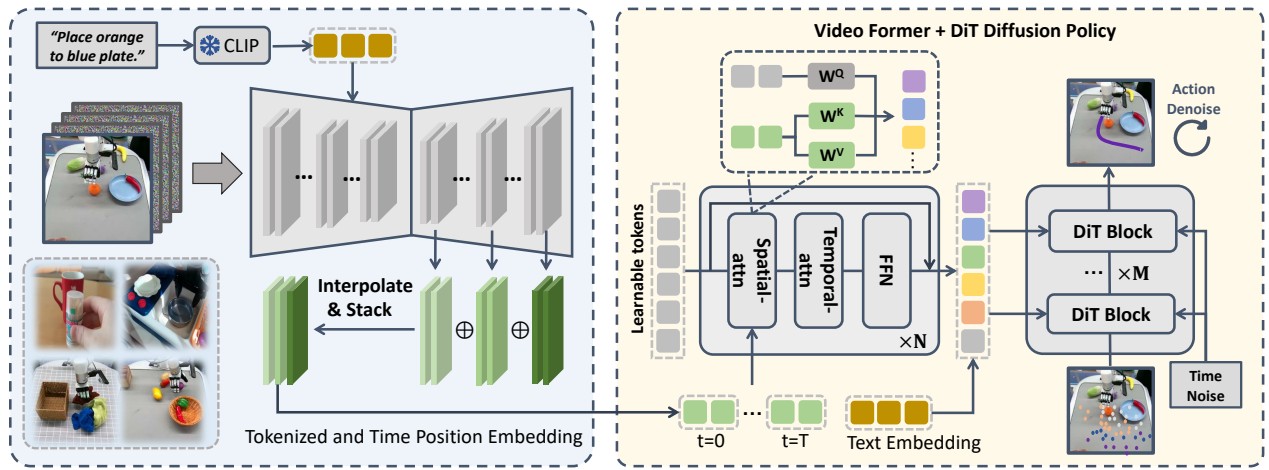

*Figure 2.* In the first stage, VPP fine-tunes a general-purpose video foundation model into a manipulation-focused Text-guided Video Prediction (TVP) model using robot and internet manipulation datasets. In the second stage, we use video-former to aggregate the representations from the TVP model during the first forward pass, followed by the diffusion policy head. This approach enables VPP to learn an implicit inverse dynamics model from the predicted future while maintaining a high control frequency.

predict future sequences based on image observations and instructions. However, denoising an entire video sequence is highly time-consuming and may lead to open-loop control issues, as discussed in (Du et al., 2024). Moreover, videos in their original pixel format often contain excessive, irrelevant information that can interfere with effective decision-making.

To address these concerns, we employ the video diffusion model primarily as a "vision encoder" rather than a "denoiser" by performing only a single forward step. Our insight is that the first forward step, while not yielding a clear video, still provides a rough trajectory of future states and valuable guidance. This insight is verified in our experiment section and shown in Fig 4. Specifically, we concatenate the current image $s_0$ with the final noised latent $q(x_{t'}|x_0)$ (typically white noise) and input this combination into the TVP model. We then directly leverage the latent features. Previous work (Xiang et al., 2023) highlights that up-sampling layers in diffusion models yield more effective representations. The feature at the $m^{th}$ up-sampling layer, with width $W_m$ and height $H_m$, is expressed as:

$$L_m = V_\theta(x_{t'}, l_{emb}, s_0)_{(m)}, L_m \in \mathbb{R}^{T \times C_m \times W_m \times H_m}$$

To effectively aggregate features from the up-sampling layers and eliminate the need for manual layer selection, we propose an automatic method for aggregating features across different layers. First, we linearly interpolate each layer's feature map to the same height and width $W_p \times H_p$:

$$L'_m = \text{Interpolation}(L_m), L'_m \in \mathbb{R}^{T \times C_m \times W_p \times H_p}$$

We then stack the features along the channel dimension. The final predictive visual representation $F_p \in$

$\mathbb{R}^{T \times (\sum_m C_m) \times W_p \times H_p}$ is given by:

$$F_p = \text{concate}((L'_0, L'_1, \ldots, L'_m), dim = 1)$$

For a robot with multiple camera views, such as a third-view and a wristed camera, we predict the future for each view independently, denoted as $F_p^{static}, F_p^{wrist}$.

**Video Former.** These predictive representations within the video diffusion model are still high-dimensional, as they express a sequence of image features. To efficiently aggregate representations across spatial, temporal, and multi-view dimensions, we design a Video Former to consolidate this information into a fixed number of tokens. The Video Former initializes learnable tokens $Q_{[0:T,0:L]}$ with fixed length $T \times L$, performing spatial-temporal attention (Blattmann et al., 2023b) on each corresponding frame, followed by feed-forward layers. Formally, this branch can be expressed as follows where $i$ is the index of frame:

$$Q' = \{\text{Spat-Attn}(Q[i], (F_p^{static}[i], F_p^{wrist}[i]))\}_{i=0}^T$$
$$Q'' = \text{FFN}(\text{Temp-Attn}(Q')). \tag{5}$$

**Action Generation.** After the Video-Former aggregates the Predictive feature into learnable tokens $Q''$, a diffusion policy is employed as the action head to generate the action sequence $a_0 \in A$ based on $Q''$. We integrate the aggregated presentation $Q''$ into diffusion transformer blocks using cross-attention layers. The diffusion policy aims to reconstruct the original actions $a_0$ from noised action $a_k = \sqrt{\bar{\beta}_k} a_0 + \sqrt{1 - \bar{\beta}_k} \epsilon$, where $\epsilon$ represents white noise, and $\bar{\beta}_k$ is the noisy coefficient at step $k$. This step can be interpreted as learning a denoiser $D_\psi$ to approximate the

| Category | Method | Annotated Data | $i^{th}$ Task Success Rate | | | | | |
|---|---|---|---|---|---|---|---|---|
| | | | 1 | 2 | 3 | 4 | 5 | Avg. Len $\uparrow$ |
| Direct Action Learning Method | RT-1 | 100%ABC | 0.533 | 0.222 | 0.094 | 0.038 | 0.013 | 0.90 |
| | Diffusion Policy | 100%ABC | 0.402 | 0.123 | 0.026 | 0.008 | 0.00 | 0.56 |
| | Robo-Flamingo | 100%ABC | 0.824 | 0.619 | 0.466 | 0.331 | 0.235 | 2.47 |
| Future Prediction Related Method | Uni-Pi | 100%ABC | 0.560 | 0.160 | 0.080 | 0.080 | 0.040 | 0.92 |
| | MDT | 100%ABC | 0.631 | 0.429 | 0.247 | 0.151 | 0.091 | 1.55 |
| | Susie | 100%ABC | 0.870 | 0.690 | 0.490 | 0.380 | 0.260 | 2.69 |
| | GR-1 | 100%ABC | 0.854 | 0.712 | 0.596 | 0.497 | 0.401 | 3.06 |
| | Vidman | 100%ABC | 0.915 | 0.764 | 0.682 | 0.592 | 0.467 | 3.42 |
| 3D Method | RoboUniview | 100%ABC | 0.942 | 0.842 | 0.734 | 0.622 | 0.507 | 3.65 |
| Ours | **VPP (ours)** | 100%ABC | **0.965** | **0.909** | **0.866** | **0.820** | **0.769** | **4.33** |
| Data Efficiency | GR-1 | 10%ABC | 0.672 | 0.371 | 0.198 | 0.108 | 0.069 | 1.41 |
| | **VPP (ours)** | 10%ABC | **0.878** | **0.746** | **0.632** | **0.540** | **0.453** | **3.25** |

*Table 1.* Zero-shot long-horizon evaluation on the Calvin ABC→D benchmark where agent is asked to complete five chained tasks sequentially based on instructions. VPP demonstrates a significant improvement in the average task completion length (Avg. Len).

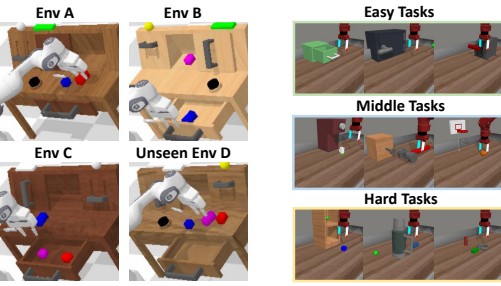

*Figure 3.* CALVIN and Metaworld benchmarks.

| Task Level (Numbers) | Easy (28 tasks) | Middle (11 tasks) | Hard (11 tasks) | Average $\uparrow$ (50 tasks) |
|---|---|---|---|---|
| RT-1 | 0.605 | 0.042 | 0.015 | 0.346 |
| Diffusion Policy | 0.442 | 0.062 | 0.095 | 0.279 |
| Susie | 0.560 | 0.196 | 0.255 | 0.410 |
| GR-1 | 0.725 | 0.327 | 0.451 | 0.574 |
| **VPP (ours)** | **0.818** | **0.493** | **0.526** | **0.682** |

*Table 2.* Multi-task success rate on Metaworld. We use a single language-conditioned policy to solve all 50 tasks.

noise $\epsilon$ and minimize the following loss function:

$$\mathcal{L}_{\text{diff}}(\psi; A) = \mathbb{E}_{a_0, \epsilon, k} \| D_\psi(a_k, l_{emb}, Q'') - a_0 \|^2 \quad (6)$$

## 5. Experiments

In this section, we conduct extensive experiments on both simulated and real-world robotic tasks to evaluate the performance of the video prediction policy (VPP). We aim to answer the following questions:

1. Can VPP achieve a higher success rate in manipulation tasks with predictive visual representations?

2. How do the video pre-training and internet manipulation datasets enhance the performance of VPP?

3. How does predictive representation compare to previous visual representations?

4. Which layer of the video diffusion model provides the most effective predictive visual representations?

### 5.1. Simulation Setups and Baselines

**CALVIN Benchmark.** CALVIN (Mees et al., 2022) is a widely used benchmark designed to assess the instruction-following capability of robotic policies in long-horizon manipulation tasks. We focus on the challenging ABC→D

setting, where the agent is trained in the ABC environment and evaluated in the unseen D environment, as illustrated in Figure 3. We use settings same as GR1 (Wu et al., 2023a) which only use the language-annotated ABC datasets for training.

**MetaWorld Benchmark.** Metaworld (Yu et al., 2020) features a Sawyer robot performing various manipulation tasks and is widely used to evaluate the precision and dexterity of robotic policies. As shown on the right of Figure 3, it includes 50 tasks with a rich array of operating objects at different levels of difficulty (Radosavovic et al., 2023). We use official Oracle policy to collect 50 trajectories for each task as our training dataset.

**VPP Training Details.** As outlined in Sec. 4, we use a two-stage training process. In the first stage, we fine-tune a video foundation model into a manipulation-focused TVP model. The videos used in this stage include 193,690 human manipulation trajectories (Goyal et al., 2017) and 179,074 robotic manipulation trajectories (O'Neill et al., 2023), along with downstream task videos, such as the official Calvin ABC videos, the MetaWorld videos, and real-world videos. Given the varying scales and quality of these datasets, we apply different sampling ratios, following the approach in

Octo (Team et al., 2024). Detailed dataset scales and sampling ratios can be found in Appendix B. Fine-tuning the video model takes 2-3 days on eight NVIDIA A100 GPUs. In the second stage, we train a generalist policy with Calvin or Metaworld dataset, which requires approximately 6-12 hours on four NVIDIA A100 GPUs.

**Policy Roll-out Details.** Previous works choose to denoise high-precision videos, a process that is time-consuming and results in low-frequency (Black et al., 2023), or even open-loop control (Du et al., 2024). In contrast, our approach uses the TVP model as an encoder rather than a denoiser, ensuring that each observation is processed through the TVP model only once, which takes less than 160 ms. Then downstream policy generate action conditioned on the predictive representation. This modification allows us to achieve a significantly higher frequency of 7-10 Hz with consumer-level NVIDIA RTX 4090 GPU. Additionally, we implement action chunking (Chi et al., 2023) with 10 steps to further improve the control frequency.

**Comparisons.** Generalist robot policy has been widely explored in previous studies. In our experiments, we opted to compare against a representative subset of prior methods that have either achieved state-of-the-art performance or share a similar approach with our methods.

- RT-1 (Brohan et al., 2022). A direct action learning robot policy that integrates semantic information using Efficient-Net with FiLM-conditioning, followed by token learners for action learning.
- Diffusion Policy (Chi et al., 2023). A direct action learning policy with novel action diffusers.
- Robo-Flamingo (Li et al., 2023). A direct action learning policy that leverages a pre-trained LLM, incorporating visual information into each layer in a flamingo style (Alayrac et al., 2022).
- Uni-Pi (Du et al., 2024). Begins by learning a video prediction model to generate future sequences and then learns an inverse kinematics model between two frames to determine actions.
- MDT (Reuss et al., 2024). Learns a diffusion transformer policy along with an auxiliary mae loss to reconstruct one masked future frame.
- Susie (Black et al., 2023). Uses a fine-tuned Instruct-Pix2Pix (Brooks et al., 2023) model to generate a goal image and learns a downstream diffusion policy conditioned on the goal image.
- GR-1 (Wu et al., 2023a). Learns video and action sequences jointly using an auto-regressive transformer. During policy execution, GR-1 outputs one future frame followed by one action.
- Robo-Uniview (Liu et al., 2024). Learns a 3d-aware visual encoder with 3d occupation loss to assist policy learning.

- Vidman (Wen et al., 2024). Pre-trained on the Open X-Embodiment dataset (OXE) video datasets and use a layer-wise self-attention adapter to transform video representation into policy model. However, Vidman did not finetune video model on down-stream tasks which lead to sub-optimal performance.

**Quantitative Results.** The comparisons on the Calvin benchmark are shown in Table 1. Results for Robo-Flamingo, Susie, GR-1, and 3D Diffuser Actors are recorded from their original papers. The MDT result is run on official implementation. The RT-1 result is sourced from (Li et al., 2023) and the Uni-Pi result from (Black et al., 2023). We also ran the Diffusion Policy based on the official open-source codebase with CLIP language conditions. Our proposed Video Prediction Policy significantly improved the previous state-of-the-art result from an average task completion length of 3.65 to 4.33. Even with only 10% of the annotated Calvin ABC data used for training, our method still achieved a length of 3.25, which exceeds the results of related methods using full data. Furthermore, the Video Prediction Policy also achieved the best performance in the MetaWorld benchmark with 50 tasks, outperforming the similar strongest GR-1 baseline by 10.8% in average success rate.

**Visualizations of Predictive Representations.** Since we use the video prediction model as a vision encoder and perform a single forward pass to obtain predictive representations, we are curious about the quality of these representations. In Figure 4 , we visualize the ground truth future, single-step predictions, and 30-step denoised predictions. We can observe that single-step representation already conveys valuable information, such as the movement of objects and the robot arm, which effectively supports downstream action learning.

### 5.2. Ablation Study

VPP achieves significant improvements in simulated experiments. In this section, we conduct ablation studies to identify the effectiveness of different components of VPP. *All ablation study are performed on Calvin ABC-D benchmark and evaluated with average task completion length.*

**Effectiveness of Predictive Visual Representations.** To verify the effectiveness of representation inside VDM, we replace the VDM vision encoder with several other pre-trained vision encoders designed for embodied tasks, while keeping all other components and settings unchanged.

1. Stable-VAE (Blattmann et al., 2023a), pre-trained with a VAE image reconstruction loss. Since the VAE encoder-decoder already performs well in reconstructing images from video datasets, we did not perform further fine-tuning. The input 256×256 images are encoded into 32×32 features with VAE, which are then

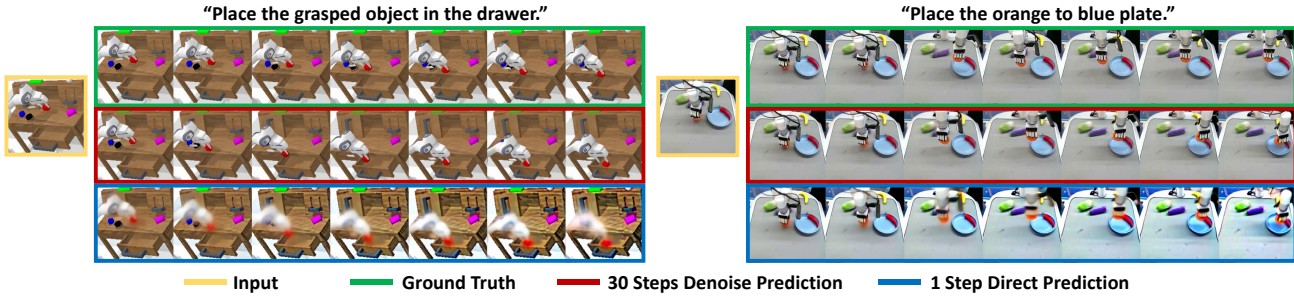

"Place the grasped object in the drawer."    "Place the orange to blue plate."

━━ **Input**    ━━ **Ground Truth**    ━━ **30 Steps Denoise Prediction**    ━━ **1 Step Direct Prediction**

*Figure 4.* Visualization of one-step forward visual representations. We can observe that one-step representation already provide valuable information on physical evolution, although the textures and details are not precise.

| Encoder | Pre-training Type | Avg. Length ↑ |
|---------|-------------------|---------------|
| VDM (ours) | Video Generation | **4.33** |
| Stable-VAE | VAE Reconstruction | 2.58 |
| VC-1 | MAE Reconstruction | 1.23 |
| Voltron | MAE Reconstruction+ Language Generation | 1.54 |

*Table 3.* Ablation study on different visual representations.

| Ablation Type | Avg. Length ↑ | Latency ↓ |
|---------------|---------------|-----------|
| VPP | **4.33** | **∼140ms** |
| VPP w/o Internet data | 3.97 | ∼140ms |
| VPP w/o Calvin video | 3.31 | ∼140ms |
| VPP w/o Internet data w/o SVD Pretrain | 1.63 | ∼140ms |
| VPP w/o Video Former | 3.86 | ∼450ms |
| VPP w/o Feature Agg. | 3.60 | ∼140ms |

*Table 4.* Ablation study on video pre-training and architecture.

resampled into 256 tokens via resampler (Jaegle et al., 2021) before passing to the diffusion policy, consistent with VPP.

2. VC-1 (Majumdar et al., 2023), pre-trained with a masked autoencoder loss. The authors note that fine-tuning vc-1 encoder with MAE loss on downstream task datasets can significantly improve performance. For a fair comparison, we first fine-tuned the model on the same video datasets used in VPP. The vc-1 features are resampled into 256 tokens with resampler and pass to policy head.

3. Voltron (Karamcheti et al., 2023), pre-trained with both MAE future reconstruction and language generation tasks. We also fine-tuned the model on our video datasets and resampled the features into 256 tokens.

The results, presented in Table 3, indicate that replacing our predictive visual representations leads to a clear decline in performance.

**Effectiveness of Video Pre-training and Internet Manipulation Datasets.** A significant advantage of the VPP is its ability to leverage the physical knowledge encoded in pre-trained video generation models and Internet manipulation datasets. We conducted experiments to verify the effectiveness of these two components. As shown in Table 4, removing the co-trained Internet manipulation data resulted in a performance decrease from 4.33 to 3.97. Further removing the pre-trained SVD model and training the video prediction model from scratch on the Calvin dataset led to a substantial performance drop. Notably, removing the video pretraining on Calvin alone also caused a significant decline.

**Effectiveness of Video Former.** The Video Former module plays a pivotal role in extracting predictive representations from the TVP model. To evaluate its effectiveness, we conduct an ablation study by removing the Video Former and directly connecting the TVP features to the diffusion policy. The results, presented in Table 5, are obtained by evaluating the complete VPP model on a single NVIDIA RTX 4090 GPU. The VPP score decreases from 4.33 to 3.86, while the inference time nearly triples. These findings indicate that the absence of the Video Former leads to a substantial degradation in both accuracy and computational efficiency compared to the full model.

**Effectiveness of Feature Aggregation Module.** Many previous works (Black et al., 2023; Wu et al., 2023a) directly use the final predicted image to learn policies. However, the image from the final layer often contains many irrelevant details that are not beneficial for the task. In contrast, we adopt a feature aggregation mechanism to leverage multiple layers of features within the up-sampling layers. We replace aggregated features with final layer features while keeping the other layers unchanged. This process lead to a decrease in the average task completion length on the Calvin benchmark, from 4.33 to 4.05. More ablations on different layers can be found at Appendix C.2.

### 5.3. Real World Experiments

We further verified the Video Prediction Policy on two real-world hardware platforms.

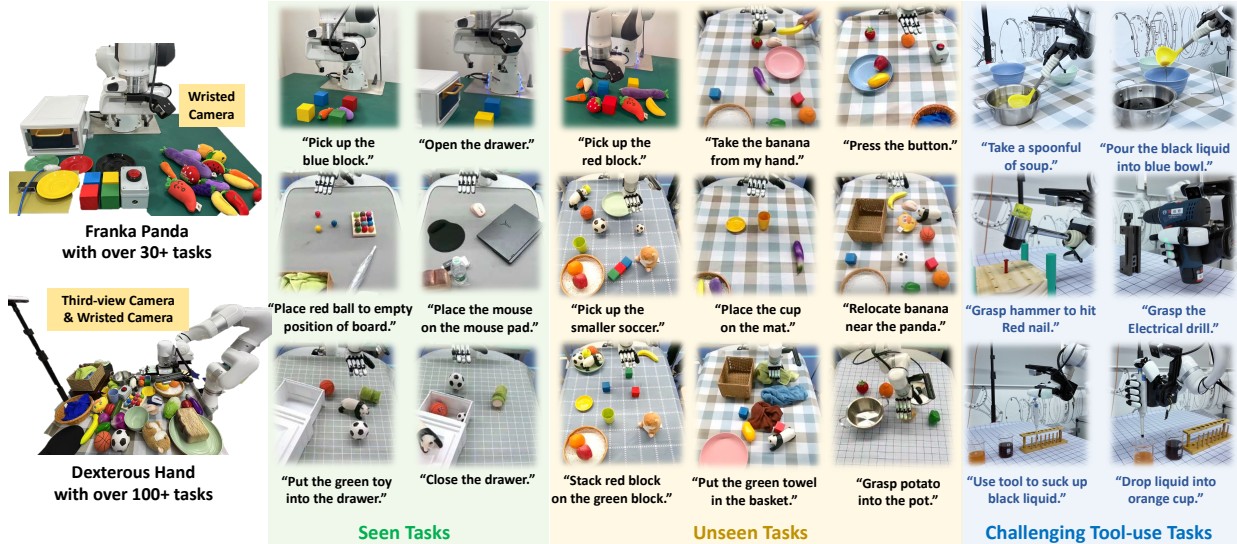

*Figure 5.* Two real-world hardware platforms and visualizations of sampled tasks. We consider a task as "unseen task" if the operated object or the background scene do not appear in training datasets.

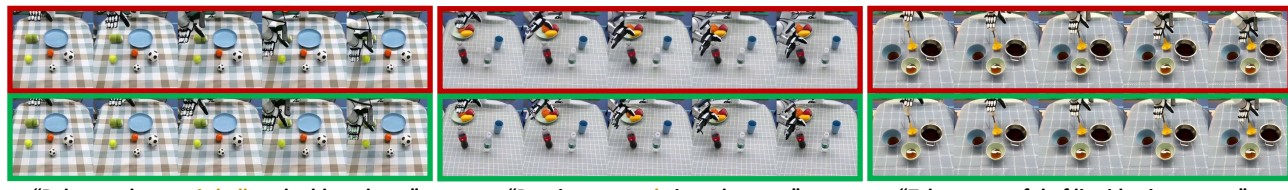

"Relocate the **tennis ball** to the blue plate."    "Pouring **coca cola** into the cup."    "Take a spoonful of liquid using **spoon**."

*Figure 6.* Predictions and executions on **unseen tasks**. Video prediction model generate reasonable futures (red). Real execution trajectories (green) is also closely aligned to the video predicted future (red).

**Franka Panda Robot Arm.** On the Franka Panda platform, we collected 2,000 trajectories for over 30 tasks in 6 categories: picking, placing, pressing, routing, opening, and closing. We divided the tasks into seen and unseen categories. A task is considered unseen if the operated object is new or the background scene is new.

**Xarm with 12-degree Xhand Dexterous Hand.** On the dexterous hand platform, we collected 4,000 trajectories over 100+ tasks in 13 categories, including picking, placing, cup-upright, relocating, stacking, passing, pressing, unplugging, opening, closing, pouring, suction, and knocking. We also define a task as unseen if the operated object is new or the background scene is new. Additionally, we included four challenging tool-use tasks, including the use of a spoon, hammer, electrical drill, and pipette for chemistry tasks. More task details can be found in Appendix A.

**Training and Rollout Details.** We employ the same text-guided video prediction (TVP) model as in our simulated experiments, trained on both internet datasets and collected real-world data. Then a generalist robot policy is learned to solve all tasks in the domain conditioned on instructions. The hardware platform and visualizations of some selected tasks are shown in Figure 5.

| **Franka Panda** | DP | Susie | GR-1 | VPP(ours) |
|---|---|---|---|---|
| Seen Tasks | 0.42 | 0.56 | 0.52 | **0.85** |
| Unseen Tasks | 0.25 | 0.46 | 0.38 | **0.73** |

| **Dexterous Hand** | DP | Susie | GR-1 | VPP(ours) |
|---|---|---|---|---|
| Seen Tasks | 0.28 | 0.45 | 0.32 | **0.75** |
| Unseen Tasks | 0.11 | 0.28 | 0.15 | **0.60** |
| Tool-use Tasks | 0.05 | 0.23 | 0.15 | **0.68** |

*Table 5.* Success rates on real-world tasks. Due to space limit, we only show the average success rate on each category. Detailed success rate can be found at Appendix A

**Quantitative Results.** Due to the complexity of deploying methods on real-world hardware, we select the strongest baseline models—GR-1, Susie, and the widely-used diffusion policy—as our baselines. For evaluation, we perform 200+ rollouts for Panda arm manipulation tasks and 500+ rollouts for dexterous hand manipulation tasks. The comparisons are in the Table 5, which indicate VPP outperforms all the baselines with a clear margin in both seen tasks, unseen tasks and tool-use tasks.

**Generalization Analysis.** we take three unseen tasks as case studies: picking up a tennis ball, pouring Coca-Cola, and using a spoon. Notably, none of these objects—tennis

ball, Coca-Cola, or spoon—appear in our collected dataset. As illustrated in Figure 6, the video prediction model forecast reasonable future states even on unseen tasks. Moreover, we observe that the actual execution trajectory closely aligns with the predicted future state. We interpret the generalization mechanism of the VPP model in two key aspects: First, video models can make correct visual predictions even on unseen tasks due to internet-scale pre-training; Second, the low-level policy learns a robust inverse dynamics model that only needs to implicitly track the movement of the robot in the predicted future, without the need to focus on new objects or backgrounds. In this way, the VPP model successfully generalizes to a wide range of unseen tasks.

## 6. Conclusion

We introduce Video Prediction Policy (VPP), a novel approach for learning a generalist robot policy. VPP learns an implicit inverse dynamics model conditioned on predictive representations inside VDMs and yields consistent improvements across both simulated and real-world tasks. As video generation models are more and more powerful these days, we aim to fully unlock the power of video model in building physical intelligence and highlight the potential of video generation models in embodied tasks.

## Acknowledgements

This research is supported by IIIS, Tsinghua University, Roboterax, UC Berkeley and Shanghai AILab. And we gratefully acknowledge the Qizhi Research Institute for providing computational resources.

## Impact Statement

This paper presents work whose goal is to advance the field of Machine Learning. There are many potential societal consequences of our work, none which we feel must be specifically highlighted here.

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

Code can be found at supplementary materiel.

## A. Real-world experiments

### A.1. Panda Maniplation

On the Franka Panda platform, we gathered demonstrations by teleoperating the Panda robotic arm using a space mouse. we collected 2k trajectories for over 30+ tasks of 6 categories including picking, placing, pressing, routing, opening, and closing. Detailed success rates for each task in seen and unseen settings are shown in Table 6.

| Seen Tasks | Diffusion Policy | Susie | GR-1 | VPP |
|---|---|---|---|---|
| Pick | 0.36 | 0.56 | 0.52 | 0.90 |
| Place | 0.40 | 0.42 | 0.38 | 0.86 |
| Press | 0.65 | 0.90 | 0.80 | 0.85 |
| Route | 0.40 | 0.55 | 0.50 | 0.75 |
| Drawer | 0.45 | 0.60 | 0.60 | 0.85 |
| Average | 0.425 | 0.563 | 0.519 | **0.856** |
| Unseen Tasks | Diffusion Policy | Susie | GR-1 | VPP |
| Pick | 0.24 | 0.40 | 0.32 | 0.80 |
| Place | 0.12 | 0.44 | 0.32 | 0.72 |
| Press | 0.50 | 0.60 | 0.60 | 0.80 |
| Route | 0.20 | 0.50 | 0.50 | 0.70 |
| Drawer | 0.40 | 0.50 | 0.40 | 0.60 |
| Average | 0.250 | 0.463 | 0.388 | **0.737** |

*Table 6.* Specific success rate at category level. In seen tasks, We evaluate pick and place tasks 50 times and other tasks 20 times respectively. In unseen tasks, we evaluate pick and place tasks 25 times and other tasks 10 times respectively

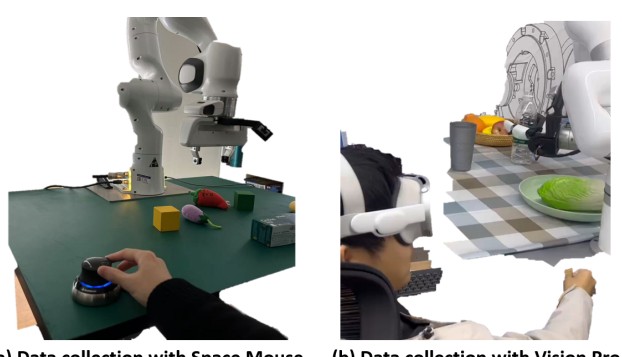

(a) Data collection with Space Mouse    (b) Data collection with Vision Pro

*Figure 7.* Data collection setups.

### A.2. Dexterous Manipulation

To collect data for dexterous manipulation, we employ Vision-Pro to capture the finger joint movements of the human hand, which are then retargeted to our 12-degree-of-freedom dexterous hand. This setup enables a human operator to directly control the dexterous hand during various manipulation tasks. We collected 4.0k trajectories over 100+ tasks of 13 categories, including picking, placing, cup-upright, relocating, stacking, passing, pressing, unplugging, opening, and closing. A low-level PD controller is used to smooth the trajectories generated by VPP.

The detailed success rates for each task category in both seen and unseen settings are shown in Table 7.

| Seen Tasks | Diffusion Policy | Susie | GR-1 | VPP |
|---|---|---|---|---|
| Pick | 0.38 | 0.61 | 0.48 | 0.83 |
| Pick&Place | 0.35 | 0.55 | 0.40 | 0.79 |
| Cup-upright | 0.00 | 0.00 | 0.00 | 0.64 |
| Relocate | 0.28 | 0.44 | 0.16 | 0.80 |
| Stack | 0.00 | 0.08 | 0.00 | 0.64 |
| Pass | 0.040 | 0.00 | 0.00 | 0.48 |
| Press | 0.68 | 0.96 | 0.64 | 0.96 |
| Unplug | 0.00 | 0.00 | 0.00 | 0.52 |
| Drawer | 0.40 | 0.64 | 0.48 | 0.72 |
| Average | 0.287 | 0.450 | 0.319 | **0.749** |
| Unseen Tasks | Diffusion Policy | Susie | GR-1 | VPP |
| Pick | 0.12 | 0.42 | 0.26 | 0.75 |
| Pick&Place | 0.08 | 0.32 | 0.20 | 0.68 |
| Cup-upright | 0.00 | 0.00 | 0.00 | 0.40 |
| Relocate | 0.12 | 0.32 | 0.12 | 0.76 |
| Stack | 0.00 | 0.00 | 0.00 | 0.56 |
| Pass | 0.00 | 0.00 | 0.00 | 0.32 |
| Press | 0.44 | 0.76 | 0.40 | 0.88 |
| Unplug | 0.00 | 0.00 | 0.00 | 0.20 |
| Drawer | 0.28 | 0.44 | 0.24 | 0.56 |
| Average | 0.110 | 0.328 | 0.159 | **0.605** |
| Tool-use Tasks | Diffusion Policy | Susie | GR-1 | VPP |
| Spoon | 0.0 | 0.4 | 0.3 | 0.9 |
| Hammer | 0.2 | 0.2 | 0.1 | 0.6 |
| Drill | 0.0 | 0.1 | 0.2 | 0.8 |
| Pipette | 0.0 | 0.0 | 0.0 | 0.4 |
| Average | 0.05 | 0.23 | 0.15 | **0.68** |

*Table 7.* Specific success rate at category level. In seen tasks, We evaluate pick and place tasks 100 times and other tasks 25 times respectively. In unseen tasks, we evaluate pick and place tasks 50 times and other tasks 20 times respectively. We evaluate each tool-use task for 10 times.

# B. Video Prediction Model

## B.1. Datasets Sample Ratios

Given the varying quality and scale of these datasets, we have introduced different sample ratios to appropriately balance the influence of different datasets, similar to (Team et al., 2024). Detailed information is shown in Table 8.

## B.2. Quantitative result on Prediction Quality

Our TVP models successfully predict future frames on validation datasets across diverse manipulation tasks, with some prediction results visualized in Appendix B.3. Additionally, we evaluate the quantitative FVD metric (Unterthiner et al., 2018) on the bridge datasets (Ebert et al., 2021), following the evaluation settings in Seer (Gu et al., 2023). The results are shown in Table 9. Surprisingly, our model easily outperforms the previous TVP model. We attribute this improvement to our use of the pre-trained video foundation model SVD (Blattmann et al., 2023a), which the earlier TVP model did not leverage, giving us a significant advantage.

| Dataset Type | Name | Trajectory Numbers | Smaple Ratio |
|---|---|---|---|
| Internet
Human Maniplation | Something-
something-v2 | 191,642 | 0.30 |
| Internet
Robot
Datasets | RT-1 | 87,212 | 0.15 |
| | Bridge | 23,377 | 0.15 |
| | BC-Z | 43,264 | 0.08 |
| | Taco-Play | 3,603 | 0.01 |
| | Jaco-Play | 1,085 | 0.01 |
| | Calvin-ABC | 18,033 | 0.10 |
| | Metaworld | 2,500 | 0.05 |
| Self-Collected
Datasets | Panda Arm | 2,000 | 0.05 |
| | Dexterous Hand | 2,476 | 0.10 |
| Total | - | 375,192 | 1.00 |

*Table 8.* We outline the dataset scales and sample ratios used for training our manipulation text-guided video prediction model. Following (Gu et al., 2023), we **exclude** 5,558 bridge trajectories and 2,048 something-something-v2 trajectories during training, reserving them for validation. For all other datasets, 3% of the trajectories are excluded and used as validation datasets.

| **Bridge** | VideoFusion | Tune-A-Video | Seer | VPP |
|---|---|---|---|---|
| **FVD**↓ | 501.2 | 515.7 | 246.3 | **41.4** |

*Table 9.* Quantitative evaluation of prediction quality on bridge datasets. The results of VideoFusion (Luo et al., 2023), Tune-A-Video (Wu et al., 2023b), Seer (Gu et al., 2023) are copied from (Gu et al., 2023).

### B.3. More Visualization of Complete Prediction Results

We present additional visualizations of prediction results from our fine-tuned manipulation TVP model. Predictions on human manipulation datasets are displayed in Figure 8, and those on robotic manipulation datasets are illustrated in Figure 10. All trajectories are sampled from the validation datasets and are predicted using the same manipulation TVP model. Each sample was denoised in 30 steps using classifier-free guidance set at 7.5, as described in (Gu et al., 2023). Our TVP model predicts a horizon of 16, and we visualize 8 frames at a skip step of 2 due to space constraints.

### B.4. More Visualizations of Predictive Representations

We visualize the intermediate predictive representations through one-step direct predictions. Additional visualizations can be found in Figure 9. As discussed in the experimental section, while the textures and details in the one-step forward videos are not precise, they still offer valuable insights into physical evolution. The movements of objects and robot arm itself already can be reflected in the visualized representations.

## C. More Details for Experiments

### C.1. Structure details

We provide the VPP architecture and hyperparameter setting details in four evaluate environments, as shown in Table 13. The transformer block in TVP follows the setting in (Blattmann et al., 2023a), and the rest of the hyperparameter in Diffusion Transformer follows the work (Reuss et al., 2024).

### C.2. More ablation

In this section, we present additional ablation experiments conducted under the ABC→D setting of CALVIN (Mees et al., 2022).

**Ablation 1 on the video former** entails the removal of the Temporal-attn module from the Video Former while maintaining all other configurations same as VPP. The results, displayed in Table 12, demonstrate that the Temporal-attn module could enhance the temporal comprehension capabilities of the Video Former.

**Ablation 2 on the number of denoising steps** introduces a 2-step denoising process in the TVP to derive the predictive

visual representation. The outcomes are summarized in Table 12, revealing that the 2-step process did not yield superior performance. We hypothesize this is because a single denoising step suffices to generate an effective representation for trajectory prediction in our configuration. Additionally, the 2-step denoising process nearly doubles the inference time and reduces the control frequency by half. Due to these factors, we opted for a one-step direct encoder in our main experiments.

**Single-view Ablation** evaluate the Calvin ABC→D task using only a single observation viewpoint (static view) and find that the success rate for Task 5 reaches 3.58. This even surpasses the success rate achieved by the state-of-the-art 3D Diffuser Actor, which utilizes two viewpoints along with depth images.

**Ablations on using different layers of features** The average task completion length are listed in Table 10.

**Ablations on using different diffusion time-step** The average task completion length are listed in Table 11.

| Calvin abc-d | Layer-3 | Layer-6 | Layer-9 | Layer-12 | VPP |
|---|---|---|---|---|---|
| Avg. Len | 3.72 | 3.88 | 4.29 | 4.05 | 4.33 |

*Table 10.* Ablations on different layers of features.

| Calvin abc-d | Time-step 10 | Time-step 20 | Time-step 30 |
|---|---|---|---|
| Avg. Len | 4.21 | 4.33 | 4.25 |

*Table 11.* Ablations on the Use of different diffusion time-step.

| Method | Tasks completed in a row | | | | | |
|---|---|---|---|---|---|---|
| | 1 | 2 | 3 | 4 | 5 | Avg. Len ↑ |
| VPP(Ours) | 0.965 | 0.909 | 0.866 | 0.820 | 0.769 | **4.33** |
| VPP(Single-view) | 0.909 | 0.815 | 0.713 | 0.620 | 0.518 | 3.58 |
| Ablation.1 | 0.949 | 0.900 | 0.839 | 0.780 | 0.714 | 4.18 |
| Ablation.2 | 0.951 | 0.904 | 0.840 | 0.777 | 0.718 | 4.19 |

*Table 12.* More ablation studies.

### C.3. Baseline Implementations

The baseline methods, including RT-1 (Brohan et al., 2022), GR-1 (Wu et al., 2023a), and Diffusion Policy (Chi et al., 2023), are implemented based on their official repositories. For comparison with Susie (Black et al., 2023) in both the Metaworld and real-world manipulation scenarios, we adopt InstructPix2Pix (Brooks et al., 2023) as the future frame predictor and use an image-goal Diffusion Policy (Chi et al., 2023) to generate the state sequence.

| Type | Name | Calvin | Metaworld | Franka Panda | Xhand |
|------|------|--------|-----------|--------------|-------|
| Prediction | Video lens | 16 | 8 | 16 | 16 |
| | Action shape | $10 * 7$ | $4 * 4$ | $10 * 7$ | $10 * 18$ |
| TVP | Language shape | $20 * 512$ | $20 * 512$ | $20 * 512$ | $20 * 512$ |
| | Image shape | $256 * 256$ | $256 * 256$ | $256 * 256$ | $256 * 256$ |
| Video Former | Token shape | $16 * 14 * 384$ | $8 * 28 * 384$ | $14 * 16 * 384$ | $14 * 16 * 384$ |
| | Input dim | 1280 | 1280 | 1280 | 1280 |
| | Latent dim | 512 | 512 | 512 | 512 |
| | Num heads | 8 | 8 | 8 | 8 |
| | num Layers | 6 | 6 | 6 | 6 |
| Diffusion Transformer | Latent dim | 384 | 384 | 384 | 384 |
| | Condition shape | $225 * 384$ | $225 * 384$ | $225 * 384$ | $225 * 384$ |
| | Num heads | 8 | 8 | 8 | 8 |
| | Encoder Layers | 4 | 4 | 4 | 4 |
| | Decoder Layers | 4 | 4 | 4 | 4 |
| | Sampling Steps | 10 | 10 | 10 | 10 |
| Hyperparameter | TVP batchsize | 4 | 4 | 4 | 4 |
| | Policy batchsize | 76 | 64 | 128 | 128 |
| | Epoch nums | 12 | 30 | 30 | 40 |
| | Learning rate | $1 * 10^{-4}$ | $5 * 10^{-5}$ | $1 * 10^{-4}$ | $1 * 10^{-4}$ |

*Table 13.* Hyper-parameters in the Video Prediction Policy (VPP).

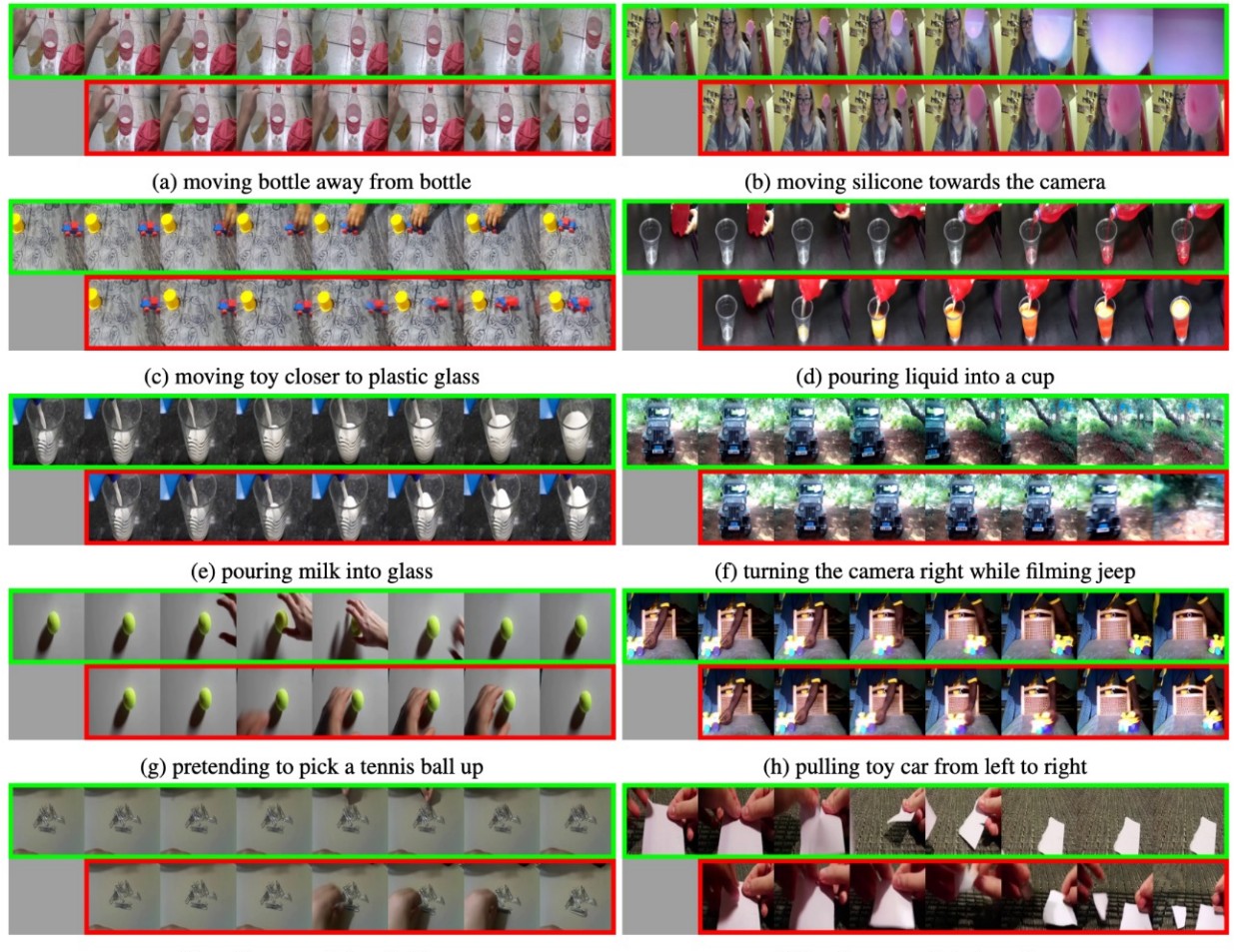

Figure 8. **Visualization of video prediction results on Internet human manipulation validation datasets with 30 steps de-noising**. The green frames indicate the ground truth while the red frames indicate the predicted futures. Zoom in for better comparisons.

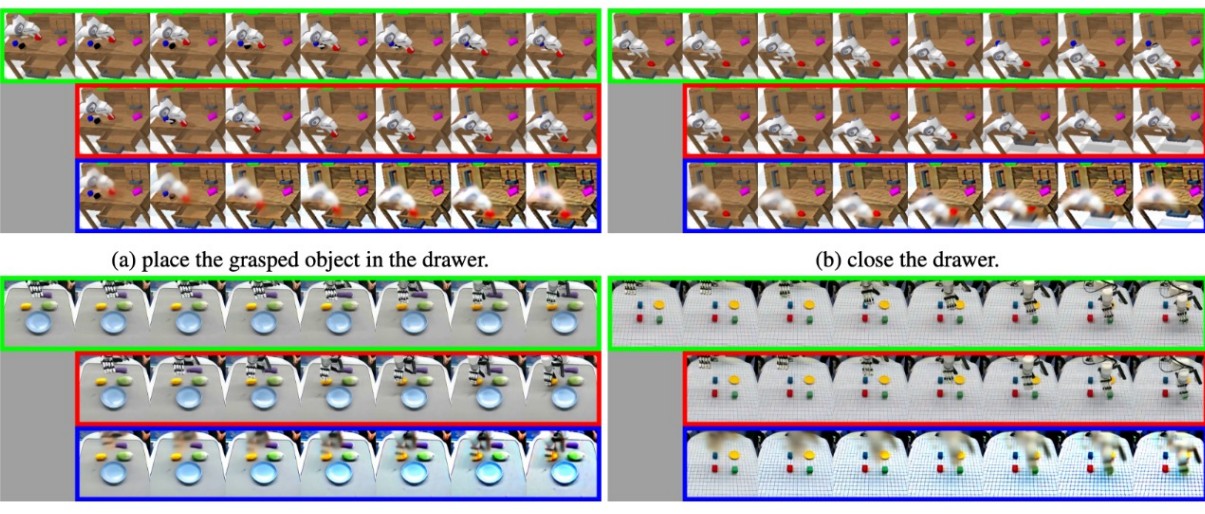

Figure 9. **Visualization of Predictive representations**. Green frames represent the ground truth, red frames correspond to the predicted future states, and blue frames illustrate the visualized predictive representations. Zoom in for better comparisons.

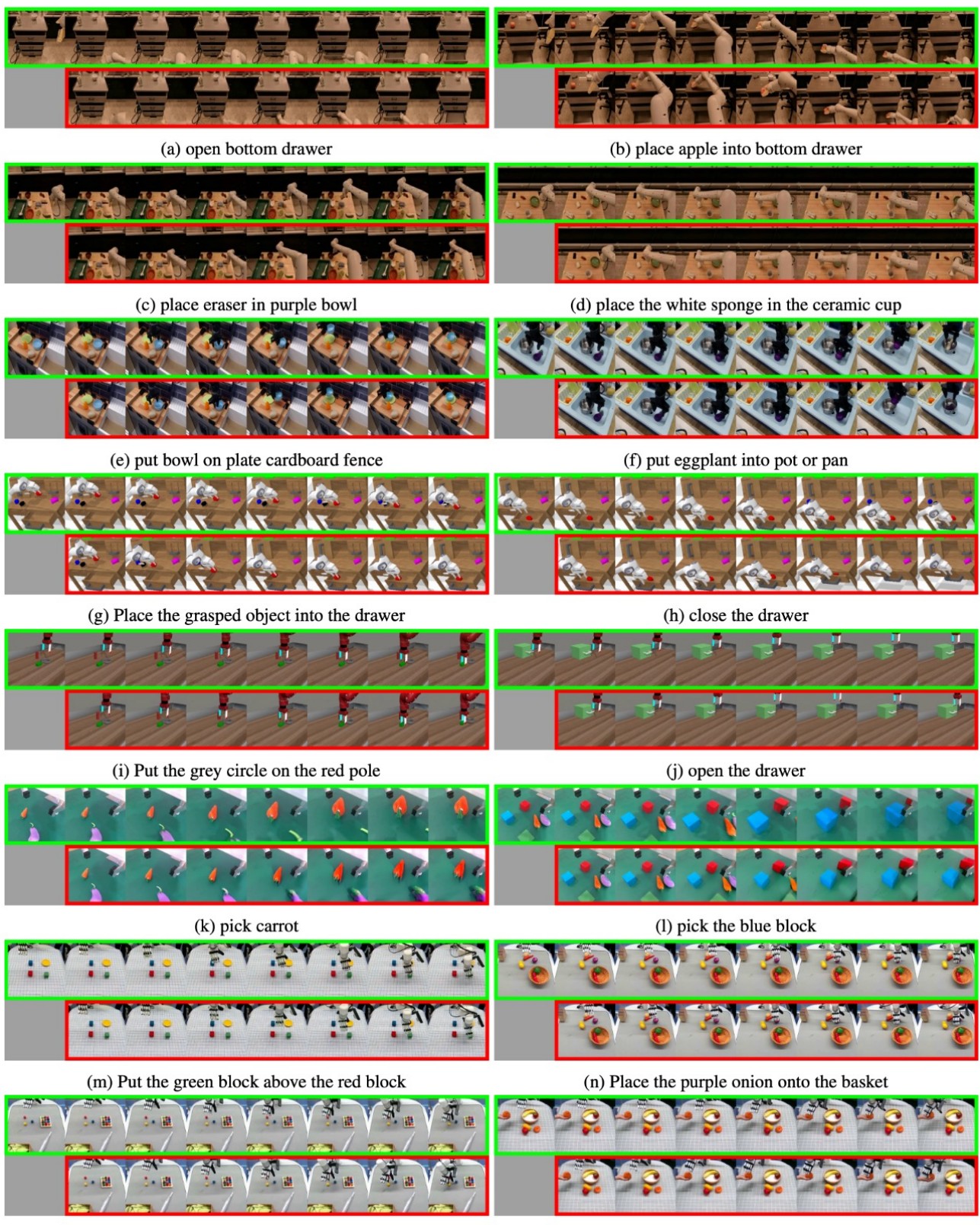

Figure 10. **Visualization of video prediction results on robotic datasets with 30 steps de-noising**. The green frames indicate the ground truth while the red frames indicate the predicted futures. (a)-(j) are sourced from internet robotic while (k)-(p) are from self-collected datasets. Zoom in for better comparisons.

