# OpenReview forum: "Video Prediction Policy: A Generalist Robot Policy with Predictive Visual Representations"
_ICML.cc/2025/Conference — ICML 2025 spotlightposter_

### Official Review · Reviewer_Y6gd · 2025-02-24

**Overall Recommendation:** 4

**Summary:**

This paper proposes to use the intermediate features of video diffusion model as a visual encoder for generalizable robot action prediction. Firstly, a pre-trained video generation model is fine-tuned on robotic dataset. Then, the latent features of the first denoising step are aggregated for down-steam action prediction. Experiments in simulator and real world proves the effectiveness of the method.

**Claims And Evidence:**

1. Features from video diffusion model could better encode the dynamic information, thus are beneficial for action prediction.
This claim is supported by the comparison with baselines and the ablation on different visual encoders.

2. Efficiency. The prediction speed is faster than previous video generation model based methods, which is proved by the control frequency.

3. Generalization. The setting of CALVIN and real-world experiments validate the generalization ability of this method.

**Essential References Not Discussed:**

No.

**Experimental Designs Or Analyses:**

This paper uses the CALVIN benchmark with ABC-to-D setting and Metaworld to validate the effectiveness of the method in simulation. Real-world experiments on two different embodiments further evaluate the proposed approach. In each setting, all methods are trained with same amount of manipulation data. Seen and unseen tasks are both evaluated to show the generalization ability.

**Methods And Evaluation Criteria:**

This paper uses two simulation environments and real-world experiments to evaluate the proposed method and baselines with same robotic data (excluding data for video model training). The metrics include successful rate and task completion length. The experiments are thorough for generalizable robot manipulation problems.

**Other Comments Or Suggestions:**

1. Please clarify more about the improvement compared to GR-1 [1]. How much of the improvement is due to the use of diffusion model?

2. I am curious about the tool-using tasks in real-world evaluation. Does the training set include tool-using tasks? Since both the text and figure show no tool-using tasks in training set (line 361, line 402). Besides, could you please show some visualization of video prediction on tool-using. The current ones only show the hand approaching the tool (Figure 6).

**Other Strengths And Weaknesses:**

### Strength

1. Leveraging the features from pre-trained video generation model to better predict dynamics is well-motivated. Visual representations suitable for robot manipulation should encode both static and dynamic information.

2. The design of using latent features instead of the denoised video improves the efficiency and robustness of action prediction.

3. The experiments are thorough. The evaluations in real world are impressive, which shows the effectiveness of VPP.

### Weakness

1. The majority of tasks belongs to pick-and-place category, which is relatively simple. What's the percentage of each task category in training and testing set.

**Questions For Authors:**

Please see above comments.

**Relation To Broader Scientific Literature:**

This approach is related to open-world manipulation. Specifically, the use of video diffusion model for robot manipulation and visual representation learning for manipulation.

**Theoretical Claims:**

Do not apply.

---

> ### Author Rebuttal · Authors · 2025-04-01
>
> We sincerely appreciate your time and effort in reviewing our paper! Thank you for your support in our work!
>
> **Q1: What's the percentage of each task category in the training and testing set?**
>
> ANS: Thank you for the questions! The specific number of evaluated trajectories can be found in Appendix A in the original paper:
>
> |           Skill Name      | Pick(Grasp)|  Pick&Place | Cup-upright |Relocate/Pass    |Stack       |Drawer       |
> | :-- | -- |-- |-- |-- |-- |-- |
> |         Test percentage(evaluate)                   | 21.1\%(100/475)|  21.1\%(100/475) | 5.26\%(25/475) |5.26\%(25/475) |5.26\%(25/475) |5.26\%(25/475) |5.26\%(25/475) |
> |           Press          | Unplug     |   Water pouring   |   Hammer     | Drill       |Pipette    | Spoon      |
> |               5.26\%(25/475)          | 5.26\%(25/475)|  5.26\%(25/475) | 5.26\%(25/475) |5.26\%(25/475) |5.26\%(25/475) |5.26\%(25/475) |5.26\%(25/475) |
>
>
> We also count the trajectory numbers of each category in the training datasets:
>
> |           Skill Name      | Pick(Grasp)|  Pick&Place | Cup-upright |Relocate/Pass    |Stack       | Drawer       |
> | :-- | -- |-- |-- |-- |-- |-- |
> |         Train percentage                   | 15.2\%|  40.5\% | 2.5\% | 7.6\% |10.1\% | 5.1\% |
> |           Press          | Unplug     |   Water pouring   |   Hammer     | Drill       |Pipette    |  Spoon      |
> |               2.5\%          | 2.5\%|  5.1\% | 1.2\% | 1.2\% | 2.5\% |3.8\% |
>
> ---
>
> **Q2: Please clarify more about the improvement compared to GR-1 [1]. How much of the improvement is due to the use of the diffusion model?**
>
> ANS: Thank you for your insightful question. To ensure a fair comparison between the diffusion-based approach and the autoregressive method, we conducted an ablation study in which we replaced the policy decoder in GR-1 with the same diffusion head used in VPP while retaining the original autoregressive video prediction component. The results on the CALVIN ABC-D benchmark are shown below:
>
> |               | Avg.Len|
> | :-- | -- |
> |    GR-1 (Auto-regressive)    | 3.06 |
> |    GR-1 (diffusion policy head)    | 3.42 |
> |    VPP (ours)   | 4.33 |
>
>  We can summarize that the performance improvement brought by VPP stems from multiple factors, including the accurate future prediction enabled by the video diffusion model, the integration of the VideoFormer architecture, and the dynamic representations learned from diverse training data in pretrained SVD.
>
> ---
>
> **Q3: I am curious about the tool-using tasks in real-world evaluation. Does the training set include tool-using tasks? Since both the text and figure show no tool-using tasks in training set (line 361, line 402).**
>
> ANS: Initially, we did not include tool-use tasks in the training dataset. We observed that VPP could generate roughly correct trajectories (e.g., approaching the tool in Figure 6) in challenging tool-use tasks but consistently failed due to the high precision control required.
>
> To address this, we collected 50–200 demonstrations for each tool-use task and fine-tuned both the video and action models. All baselines were also trained with the newly collected demonstrations.
> After fine-tuning, our model achieved a higher success rate in tool-use tasks. For your convenience, we have visualized more video prediction results before/after finetuning on our website: https://sites.google.com/view/vpp-rebuttal
>
> ---
>
> **Q4: Besides, could you please show some visualization of video prediction on tool-using. The current ones only show the hand approaching the tool (Figure 6).**
>
> ANS: Yes! We have also visualized the more video prediction results for tool tasks at different stages on our website: https://sites.google.com/view/vpp-rebuttal
>
> ---
> Thank you once again for endorsing our work! We will continue refining it to meet the highest standards.

---

> > ### Comment · Reviewer_Y6gd · 2025-04-03
> >
> > Thank the authors for the rebuttal. It addresses most of my concerns.

---

> > > ### Author Response · Authors · 2025-04-04
> > >
> > > Dear Reviewer Y6gd:
> > >
> > > Thank you again for your time and effort! We really appreciate your support in our work!
> > >
> > > Best Regards,
> > >
> > > The Authors

---

### Official Review · Reviewer_uxTn · 2025-03-04

**Overall Recommendation:** 4

**Summary:**

This paper introduce Video Prediction Policy(VPP), a robotics framework that leverages video diffusion model to capture the dynamic presentation vital to policy training. VPP consists of a two-stage process: i) a general video model is fine-tuned into a text-guided video prediction (TVP) model using large-scale human and robotic manipulation datasets, ii) a robotic policy is trained on predictive representations from the TVP model, enabling implicit inverse dynamics learning.
VPP is evaluated on both simulate and real-world environments against mainstream baselines. The comparison shows that VPP i) achieves a higher success rate in manipulation tasks ii) enable generalize in unseen tasks.

## update after rebuttal
I confirm my score. Authors addressed comments and added clarity and results to the original submission.

**Claims And Evidence:**

- In the introduction part, the paper suggests that VPP requires few demonstrations to align the robot's action space with visual features, but the extent of this data efficiency is not well quantified.
- While the results suggest that VDM-based representations capture useful motion dynamics, the paper does not explicitly demonstrate how these representations encode physical laws.

**Essential References Not Discussed:**

No

**Experimental Designs Or Analyses:**

The paper claims that VPP requires few demonstrations, but it does not test performance under limited-data conditions.

**Methods And Evaluation Criteria:**

VPP and evaluation crieria are generally appropriate and well-aligned with the problem of learning generalist robotic policies. VPP is evaluated in Calvin ABC-D, metaworld, and real-world robot arm and dexterous hand, which show its ability to generalize and do complex tasks.

**Other Comments Or Suggestions:**

Some typos:
Introduction part: “Since we direct use the internal representation and avoid the need for multiple denoising steps as in previous work”, “direct” should be “directly”

**Other Strengths And Weaknesses:**

**Strengths:**
- VPP achieves good performance in both simulation and real-world tasks by learning an implicit inverse dynamics model on future representations predicted by VDMs, and avoids the need for multiple denoising steps as in previous work, so it can run in closed-loop at high frequency.
- Able to leverage physical knowledge from pre-trained video generation models and Internet manipulation datasets.

**Weakness:**
- VPP's performance is highly dependent on the quality and predictive power of the video diffusion model. If the video diffusion model cannot accurately predict future scenes, VPP's performance may be limited.
- Fine-tuning the video diffusion model requires a lot of computing resources. Fine-tuning the video model takes 2-3 days on eight NVIDIA A100 GPUs.
- It may still be difficult to accurately capture the complex physical dynamics over long time scales.

**Questions For Authors:**

Does the performance increase with the number of denoising steps increase?

**Relation To Broader Scientific Literature:**

The main contribution of this paper is the Video Prediction Policy (VPP), a new approach to learning general robotic policies using predicted visual representations inside video diffusion models (VDMs). VPP achieves significant performance gains in both simulation and real-world tasks by learning an implicit inverse dynamics model on future representations predicted by VDMs.

**Theoretical Claims:**

The theoretical foundation of VPP is not formally established through rigorous proofs or mathematical derivations.

---

> ### Author Rebuttal · Authors · 2025-04-01
>
> We sincerely appreciate your time and efforts in reviewing our paper! Thank you for your support on our work!
>
>
> **Q1: Regarding the data efficiency of VPP model**
>
> ANS: Thank you for your constructive question! In Table 1 of the original paper, we conducted experiments on the CALVIN benchmark using only 10\% of the standard dataset. Even with this limited data, VPP achieved performance comparable to previous SOTA models trained on the full dataset. These experiments demonstrate the data efficiency of VPP compared to previous methods, which stems from its powerful video pretraining process. Below, we provide additional comparisons with other methods:
>
>
> |     CALVIN ABC-D  | Data percentage|  Avg. Len |
> | :-- | -- |-- |
> |      GR-1          |   10\%   | 1.41 |
> |      GR-1          |   100\%  | 3.06 |
> | 3d-diffuser actor  | 100\%    |  3.35|
> |      CLOVER        |   100\%  | 3.53 |
> |Vidman (Concurrent*)|  100\%   | 3.42 |
> |    VPP (ours)      | 10\%     | 3.25 |
> |    VPP (ours)      | 100\%    | 4.33 |
>
>
> ---
>
> **Q2: While the results suggest that VDM-based representations capture useful motion dynamics, the paper does not explicitly demonstrate how these representations encode physical laws.**
>
> ANS: Thank you for your insightful question! To directly illustrate the physical dynamics encoded in the VDM representation, we visualize the one-step denoised VDM features in Figures 4 and 9 of the original paper. These visualizations show that the learned representations consistently maintain physical consistency and align with intuitive physical laws.
>
> For example, when a block is grasped, it moves along with the robot arm, and the color of the poured liquid matches the liquid in the bottle. We hope this analysis clarifies our claims.
>
> ---
>
> **Q3: VPP's performance is highly dependent on the quality and predictive power of the video diffusion model. If the video diffusion model cannot accurately predict future scenes, VPP's performance may be limited.**
>
> ANS: Thank you for your insightful comment—we agree with your observation! We have also noted that policy performance strongly correlates with video prediction quality. However, from another perspective, we believe **this dependency can be a strength**. As pretrained video models continue to improve, their enhanced prediction capabilities allow us to transfer generalization abilities learned from internet-scale data to embodied tasks, which is really a good thing in the data-limited robotics domain.
>
> ---
>
> **Q4: Fine-tuning the video diffusion model requires a lot of computing resources, taking 2-3 days on eight NVIDIA A100 GPUs.**
>
> ANS: Since the pretrained SVD model has 1.5 billion parameters, we use an A100 node for fine-tuning. However, by employing parameter-efficient techniques such as LoRA, we can further reduce the computational demand, which is also implemented in the code provided in the supplementary materials.
>
> ---
>
> **Q5: It may still be difficult to accurately capture the complex physical dynamics over long time scales.**
>
> ANS: Thank you for the insightful comment! As pretrained video models continue to improve, we believe their ability to predict complex dynamics will also improve. Furthermore, the VPP framework does not require long-horizon predictions, as the video model can perform high-frequency replanning (re-prediction). As described in Section 4.1, we predict 16 frames with a 0.2s time interval, resulting in a short prediction horizon of 3.2 seconds.
>
> ---
>
> **Q6: Does the performance increase with the number of denoising steps?**
>
> ANS: This is a great question! The number of denoising steps presents a trade-off: while more steps can lead to higher-quality RGB images, they also result in a much lower control frequency. We explored this in an ablation study presented in Table 10 (Ablation.2) of the original paper. Our findings show that using two denoising steps yields similar performance to a single-step approach but at nearly half the control frequency.
>
> To balance performance and efficiency, we ultimately adopted the one-step direct encoder in our main experiments. We did not experiment with higher denoising steps, as they would further reduce control frequency, making the policy impractical for real-world deployment.
>
>
> ---
> Thank you again for your endorsement in our work! We will keep polishing the work to make it meet the highest standard!

---

> > ### Comment · Reviewer_uxTn · 2025-04-03
> >
> > Thank you for addressing my and other reviewers' comments.

---

> > > ### Author Response · Authors · 2025-04-03
> > >
> > > Dear Reviewer uxTn:
> > >
> > > We are delighted to receive your response! Thank you again for your valuable time in reviewing our paper!
> > >
> > > Best Regards,
> > >
> > > Authors

---

### Official Review · Reviewer_6CJo · 2025-03-13

**Overall Recommendation:** 4

**Summary:**

The paper is about fine-tuning a pretrained image to video diffusion model on video-caption datasets that focus on object manipulations. The image to video model is turned into an image plus text to video model through the fine-tuning. This diffusion model is then used as a feature extractor (no iterative de-noising) to obtain future frame representations for a given robotics task (defined by start frame and textual command). These future representations, tagged predictive visual representations, are used as inputs to train a separate diffusion policy model on various robotics tasks. The policy model is trained separately on ground-truth robot trajectories, generating action sequences given starting image and textual command. The two detached training phases allow the diffusion model to be trained on a large-scale with weak supervision, while the policy network is learned at a task level under full supervision. Experiments on two synthetic and two real-world benchmarks demonstrate the improved performance achieved by this method over existing work.

## update after rebuttal

The authors have successfully addressed the concerns.

**Claims And Evidence:**

Yes. The key claim is that a video generation diffusion model, adequately fine-tuned, can provide representations useful for robotics downstream tasks. Extensive experimentation (on synthetic and real-world benchmarks) is provided as evidence to demonstrate usefulness of such representations.

**Essential References Not Discussed:**

N/A

**Experimental Designs Or Analyses:**

Follows standard evaluation settings from prior work.

**Methods And Evaluation Criteria:**

Yes. The method of using future prediction is well motivated (and is well explored in concurrent literature). Evaluation criteria follows standard settings and aligns with results in prior works.

**Other Comments Or Suggestions:**

Proposing weak accept given the several minor concerns listed in weaknesses above.

**Other Strengths And Weaknesses:**

Strengths

* Interesting idea of using future predictions in representation space with efficient, non-iterative feature extraction from a video diffusion model.

* Clear presentation of methodology: the representation learning as well as fine-tuning stages are explained well with good use of figures for explanation.

* Clear experimental setup following established benchmarks from prior work. Easy to observe the improvements from video diffusion representations for robotics tasks.

Weaknesses

1. Details on method

    * What diffusion time-step is used to extract TVP features during inference?

2. Additional ablations

    * In Table 4, do you have results with SVD pretrain, but only internet data training? I.e. the impact of SVD pre-training for the robotics tasks.

3. Inference speed comparison missing

    * Please add table comparing against inference speed of baselines

**Questions For Authors:**

N/A

**Relation To Broader Scientific Literature:**

Highly relevant to several recent works which explore similar settings. This work could serve as a strong motivation for various future works.

**Theoretical Claims:**

N/A

---

> ### Author Rebuttal · Authors · 2025-04-01
>
> We sincerely appreciate your time and efforts in reviewing our paper! Based on your review, we added a detailed discussion and additional experiments.
>
>
> **Q1: About the details on method: what diffusion time-step is used to extract TVP features during inference?**
>
> Thank you for your question. In our pipeline, we set the SVD noise scale and diffusion timestep to 20. We conducted preliminary experiments with diffusion timesteps of 10, 20, and 30 and found that 20 yielded slightly better performance. However, the overall performance of VPP is not sensitive to the choice of timestep. The results for different timesteps are shown below:
>
> |   CALVIN ABC-D       | Avg.Len|
> | :-- | -- |
> |    VPP time-step 10  | 4.21  |
> |    VPP time-step 20  | 4.33  |
> |    VPP time-step 30  | 4.25 |
>
> ---
>
> **Q2: In Table 4, do you have results with SVD pretrain, but only internet data training? I.e. the impact of SVD pre-training for the robotics tasks.**
>
> ANS: Thank you for the insightful question! Following your suggestion, we fine-tuned the SVD model using only internet data, without incorporating downstream robotics data. As shown below, the results indicate a clear performance decline. We believe this is because **video prediction quality plays a crucial role in action learning**. In the VPP framework, fine-tuning the video model on robot datasets enhances video prediction quality within the specific domain, allowing it to better capture the dynamics of robotic data and potentially improve performance.
>
>
>
> |               | Avg.Len|
> | :-- | -- |
> |    VPP w/o internet data    | 3.97  |
> |    VPP w/o down-stream robot dataset (newly added)   | 3.31  |
> |    VPP   | 4.33 |
>
>
> ---
>
> **Q3: Please add table comparing against inference speed of baselines.**
>
> ANS: Thank you for the constructive suggestion! Follow your suggestion, We add a comparison of the inference speed here. All inference time is evaluated on single NVIDIA 4090 GPU and average on 100 runs. We can notice that VPP achieves the best performance while keeping high frequency. Other  method containing video/imgae diffusion (e.g., Susie/Uni-Pi) requires long time to denoise a complete video.
>
> |     CALVIN ABC-D  | inference Time|  Avg. Len |
> | :-- | -- |-- |
> |    Diffusion policy      | ~100ms |  0.56 |
> |    3d-diffuser actor     | ~600ms |  3.35 |
> |    Susie    |  ~5100ms | 2.69 |
> |    GR-1     |  ~90ms   | 3.06 |
> |    MDT      |  ~110ms  | 1.55 |
> |    Uni-Pi   |  ~5500ms  | 0.92 |
> |    VPP      |  ~140ms  | 4.33|
>
> ---
>
> Thank you again for your time and effort in reviewing our work! We hope our clarification can solve all your concerns, and we are always ready to answer any further questions!

---

> > ### Comment · Reviewer_6CJo · 2025-04-07
> >
> > Most of the concerns are addressed in the rebuttal.

---

> > > ### Author Response · Authors · 2025-04-08
> > >
> > > Dear Reviewer 6CJo:
> > >
> > > We sincerely appreciate your valuable comments and kind support. We will carefully consider your suggestions and revise the paper accordingly. Thank you once again for taking the time to review our manuscript！
> > >
> > > Best Regards,
> > >
> > > Authors

---

### Official Review · Reviewer_xoHf · 2025-03-14

**Overall Recommendation:** 3

**Summary:**

The paper introduces the Video Prediction Policy (VPP), a versatile robotic policy that enhances robot control by utilizing predictive visual representations generated by text-guided video prediction models (TVPs). VPP employs a two-stage methodology: first, fine-tuning a text-guided video prediction model on manipulation datasets, and second, integrating these predictive representations into a diffusion policy. The results show that VPP surpasses existing methods across both simulated and real-world benchmarks, achieving significant gains in task success rates.

**Claims And Evidence:**

The claim regarding improved control frequency may be subject to scrutiny. For methods such as UniPi, which employ 'Video Prediction + Inverse Dynamics Model (IDM)', although multiple denoising steps are required to generate a full video sequence for enhanced downstream control, they typically only need to generate the video once (i.e., a video plan spanning a relatively long horizon). Subsequent multi-step closed-loop rollouts rely solely on the lightweight IDM, which can operate very efficiently. In contrast, VPP appears to invoke the large TVP model at every control step, potentially incurring higher computational costs. To substantiate claims about improved efficiency, evaluations focusing on the time consumed for task completion would provide a more meaningful and supportive metric.

**Essential References Not Discussed:**

A notable similarity exists between the proposed method and VidMan [1], as both approaches involve fine-tuning video diffusion models using robotic data and leveraging features derived from the denoising process, integrated with cross-attention mechanisms, for diffusion-based action prediction. Furthermore, the core motivation of this paper—leveraging predictive visual representations from video diffusion models for robotic tasks—has also been extensively explored and discussed in prior literature [2].


[1] Wen, Youpeng, et al. "Vidman: Exploiting implicit dynamics from video diffusion model for effective robot manipulation." NeurIPS 2024
[2] Xiao, Zeqi, et al. "Video diffusion models are training-free motion interpreter and controller." arXiv preprint arXiv:2405.14864.

**Experimental Designs Or Analyses:**

1. The analysis provided in Table 3 lacks persuasiveness and cannot be regarded as a fair comparison on the 'Pre-training Type'. It is noteworthy that the largest model in Voltron is a ViT-Base architecture with approximately 83 million parameters, while the Stable Video Diffusion (SVD) model employed in the proposed method exceeds 1.5 billion parameters. This substantial discrepancy in model scale undermines the validity of the comparison and calls into question the robustness of the findings.
2. '3D Diffuser Actor' is no longer the state-of-the-art method in the CALVIN benchmark. Further inclusion and comparison with more advanced methods will highlight the superiority of the proposed method.

**Methods And Evaluation Criteria:**

Leveraging inherent dynamics within video diffusion models is a promising approach for robotic control.
The evaluation framework in this paper encompasses CALVIN, MetaWorld, and real-world assessments involving both gripper and dexterous hands, providing a comprehensive evaluation of the proposed method.

**Other Comments Or Suggestions:**

Is line.117, 'we observe that the open-sourced SVD', "W" should be capitalized.

**Other Strengths And Weaknesses:**

The evaluations with the dexterous hands are barely seen in recent literature, which is valuable for validating the proposed method.

**Questions For Authors:**

1. For the second stage of training as described in the paper, are the video former and DiT policy trained in an end-to-end manner with the TVP-based backbone, or can the TVP be frozen during this stage? Clarifying this would provide insight into the flexibility and efficiency of the training process.

2. It is worth exploring whether the TVP-based backbone can generalize to unseen environments. In the current version, the TVP is extensively trained on specific downstream datasets (i.e., CALVIN and MetaWorld). How well does it generalize to novel scenarios or different robotic embodiments? The authors could consider evaluating VPP on CALVIN by training it exclusively on CALVIN data during the second stage.

3. What is the prediction horizon (i.e., the time duration of predicted videos) of the TVP? Should this horizon be adjusted based on the control frequency of the system? To what extent will it affect the downstream performance?

4. The feature aggregation process appears to have a significant impact on performance, as evidenced by the performance drop on CALVIN from 4.33 to 3.6. It would be valuable to investigate which layer's representations provide the most informative features for robotic control.

**Relation To Broader Scientific Literature:**

Discussed in the 'Essential References' section.

**Theoretical Claims:**

No theoretical claims are made in the paper.

---

> ### Author Rebuttal · Authors · 2025-04-01
>
> We sincerely appreciate your time and efforts in reviewing our paper! Based on your review, we added detailed discussions and additional experiments:
>
>
> **Q1: Control frequency comparisons to UniPi**
>
> ANS: We are afraid misunderstandings exist on the UniPi work. In page 5, sec 3.2 of the original UniPi paper, the authors state: "we use an **open-loop** controller in all our experiments in this paper."The IDM model takes two predicted images as input without any feedback from the environment, making it an open-loop control system. In contrast, VPP can replan in close-loop based on current observation at high frequency(7-10Hz) by direct using representations in the first forward pass.
>
> ---
>
> **Q2: About the comparison on Table 3: Voltron is only 83M while SVD is 1.5B**
>
> ANS: We respectfully offer a different perspective on this comparison: Previous vision encoders for embodied control were trained with specifically designed objectives on small networks; Video diffusion models with billions of parameters are pretrained on large-scale internet datasets—why not leverage their power? Beside, pretraining these vision encoder with billions of parameters from scratch requires enormous computational resources, which we really can not afford. We hope this clarification can addresses your concern!
>
> ---
>
> **Q3: About the SOTA on CALVIN benchmark**
>
> ANS: We list several works that achieve strong performance on CALVIN, including some concurrent papers. Even considering these, to the best of our knowledge, VPP remains the SOTA method on CALVIN at the time of submission.
>
> |     CALVIN ABC-D  | Release Time|  Avg. Len |
> | :-- | -- |-- |
> |    3d-diffuser actor     | Feb. 16, 2024 |  3.35 |
> |      RoboUniview     |   Jun. 27, 2024  | 3.64 |
> |      CLOVER     |   Sep. 13, 2024  | 3.53 |
> |    Vidman (Concurrent*)      |  Nov.14, 2024   | 3.42|
> |    VPP      |   | 4.33|
>
> *In ICML guidelines, papers released within 4 months to the submission DDL are concurrent and author are not required to compare to them.
>
> ---
>
> **Q4: Comparison and differences to Vidman**
>
> ANS: (1) Performance: VPP significantly outperforms Vidman, even as a concurrent work. (2) Training strategy: Vidman does not fine-tune its video model on the downstream domain, likely leading to worse video predictions and lower performance. (3) Architecture: Vidman directly uses action tokens to attend to numerous video model tokens, whereas VPP employs a VideoFormer to aggregate tokens, assisting action learning.
>
> Ablations confirm that removing downstream fine-tuning and VideoFormer both degrade performance:
>
> |               | Avg.Len|
> | :-- | -- |
> |    Vidman    | 3.42  |
> |    VPP w/o video finetuned in domain    | 3.31 |
> |    VPP w/o video former   | 3.86 |
> |    VPP   | 4.33 |
>
> ---
>
> **Q5: Comparisons to the paper "Video diffusion models are training-free motion interpreter and controller.**
>
> ANS: We carefully reviewed this paper and found that it focuses on video motion generation, which is fundamentally different from robotics tasks. While it explores controlling motion generation via video latent, our work emphasizes the predictive capability of video representations, which is crucial for embodied AI.
>
> ---
>
> **Q6: How well does video model generalize to different robotic embodiments? Consider evaluating VPP on CALVIN by training TVP model exclusively on CALVIN data**
>
> ANS: Thank you for the insightful question! As mentioned in Q4, removing CALVIN video in TVP fine-tuning reduces performance. However, in this setting, representation inside TVP models still outperform the representations from static SVD VAE model. We argue that fine-tuning the TVP model on collected robot data better utilizes the available robotic data.
>
> |               | Avg.Len|
> | :-- | -- |
> |    SVD's VAE    | 2.58 |
> |    VPP w/o video FT   | 3.31 |
> |    VPP   | 4.33 |
>
> ---
>
> **Q7: Is video model freeze in the second stage of training?**
>
> ANS: Yes! We will make it more clear in the method section.
>
> ---
>
> **Q8: Prediction horizon of TVP and affection on the downstream tasks?**
>
> ANS: Typically we set the prediction horizon longer than the action horizon. In Section 4.3, VPP predicts 16 video frames at 0.2s intervals while outputting 10 actions at 0.1s intervals. Ablating to a 0.1s × 16 frame prediction had only a slight effect on performance (4.33 → 4.21).
>
> ---
>
> **Q9: Which layer's representations provide the most informative features.**
>
> ANS: Thank you for the insightful suggestion! We perform additional ablation study by conditioning policy on different layers inside up-sampling block. The results shows that the most informative ones lie in the middle of the SVD model. Our feature aggregation mechanism avoids manual selection while achieving the best results.
>
> |               | Avg.Len|
> | :-- |  --   |
> |    Layer-3    | 3.72  |
> |    Layer-6    | 3.88  |
> |    Layer-9    | 4.29  |
> |    Layer-12   | 4.05  |
> |    VPP   | 4.33 |
>
> ---
>
> Thank you again for your time and effort in reviewing our work!

---

> > ### Comment · Reviewer_xoHf · 2025-04-06
> >
> > The detailed rebuttal and additional results are appreciated. I'd like to raise my socre accordingly.

---

> > > ### Author Response · Authors · 2025-04-06
> > >
> > > Dear Reviewer xoHf:
> > >
> > > Thank you for your review and feedback! We truly appreciate your time and effort, and we will carefully consider your suggestions in future revisions of our paper!
> > >
> > > Best Regards,
> > >
> > > Authors

---

### Decision · Program_Chairs · 2025-05-01

**Decision:**

Accept (spotlight poster)

**Comment:**

This work introduces Video Prediction Policy (VPP), which learns implicit inverse dynamics model conditioned on predicted future representations inside Video Diffusion Models (VDMs). VPP enhances robot control by utilizing predictive visual representations generated by text-guided video prediction models (TVPs). VPP consists of a two-stage process: (i) a general video model is fine-tuned into a text-guided video prediction (TVP) model using large-scale human and robotic manipulation datasets, (ii) a robotic policy is trained on predictive representations from the TVP model, enabling implicit inverse dynamics learning. VPP is evaluated on both simulated and real-world environments against mainstream baselines. The evaluation shows that the representations learned by VPP are effective for downstream robotics tasks (i) achieving a higher success rate in manipulation tasks (ii) enabling generalization in unseen tasks (iii) being more efficient (control frequency) than previous video generation based methods.

All reviewers agree that the proposed method of using future prediction is well motivated and its a promising approach for robotic control. The experiments are thorough and the evaluation criteria follows standard settings and aligns with results in prior works. The paper is well written with a clear presentation of methodology: the representation learning as well as fine-tuning stages are explained well with good use of figures for explanation. This work could serve as a strong motivation for various future works.

The authors have appropriately addressed all the concerns raised by reviewers. All additional results presented during the rebuttal should be included in the final version of the paper.